# Design, Synthesis and Preliminary Biological Evaluation of Benzylsulfone Coumarin Derivatives as Anti-Cancer Agents

**DOI:** 10.3390/molecules24224034

**Published:** 2019-11-07

**Authors:** Tao Wang, Tao Peng, Xiaoxue Wen, Gang Wang, Yunbo Sun, Shuchen Liu, Shouguo Zhang, Lin Wang

**Affiliations:** 1College of Life Science and Bio-engineering, Beijing University of Technology, Beijing 100124, China; 2Beijing Institute of Radiation Medicine, Beijing 100850, China

**Keywords:** benzylsulfone, coumarins, phosphoinositide 3-kinase (PI3K), anticancer, cell migration, molecular docking

## Abstract

In this work, a series of benzylsulfone coumarin derivatives **5a**–**5o** were synthesized and characterized. Kinase inhibitory activity assay indicated that most of the compounds showed considerable activity against PI3K. Anti-tumor activity studies of the active compounds were also carried out in vitro on the Hela, HepG2, H1299, HCT-116, and MCF-7 tumor cell lines by MTS assay. The structure–activity relationships (SARs) of these compounds were analyzed in detail. Compound **5h** exhibited the most potent activities against the mentioned cell lines with IC_50_ values ranging from 18.12 to 32.60 μM, followed by **5m** with IC_50_ values of 29.30–42.14 μM. Furthermore, **5h** and **5m** clearly retarded the migration of Hela cells in vitro. Next, an in silico molecular docking study was conducted to evaluate the binding models of **5h** and **5m** towards PI3Kα and PI3Kβ. Collectively, the above findings suggested that compounds **5h** and **5m** might be promising PI3K inhibitors deserving further investigation for cancer treatment.

## 1. Introduction

The phosphoinositide 3-kinase (PI3K) pathway, one of the most frequently altered pathways in cancer, plays a momentous role in tumorigenesis and many other cellular processes [1,2,3,4]. Considering its contributions to the regulation of cell growth, survival and metastasis, targeting major molecules within this pathway represents a pivotal opportunity for cancer therapy [5,6,7]. As one key effector node, the intracellular lipid kinases PI3Ks have been studied thoroughly and validated as promising anticancer targets over the last 30 years [8,9]. Based on their structure and function, PI3K enzymes can be divided into three classes, of which class I has been the most studied, with the subdivisions of PI3Kα, PI3Kβ, PI3Kγ and PI3Kδ [10,11]. PI3Kα and PI3Kβ are universally expressed in all cells and tissues, whereas gamma and delta isoforms are exclusively enriched in immune cells [12,13].

In recent years, small molecule inhibitors of PI3K, including pan- and isoform-specific, are currently being evaluated and have exhibited significant anti-tumor activities in laboratory and clinic, either monotherapy or in combination with cytotoxic agents [14,15,16]. To our knowledge, pan-PI3K inhibitors include but are not limited to buparlisib [17], pictilisib [18] and pilaralisib [19], targeting all isoforms. Conversely, taselisib [20], alpelisib [21], and idelalisib [22] belong to isoform-specific inhibitors.

Rigosertib (Figure 1), also known as Estybon and ON01910.Na, is a synthetic non-ATP competitive, multi-kinase inhibitor and an anticancer agent developed by Onconova Therapeutics Inc. [23]. Studies demonstrated that rigosertib evidently inhibited PI3Kα and PI3Kβ isoforms between concentrations of 1 and 10 μM in vitro, yet showed moderate inhibition of γ and δ isoforms at high concentrations above 100 μM [24]. Furthermore, rigosertib induced cytotoxicity against varieties of tumor cell lines with IC_50_ (half maximal inhibitory concentration) values in the nanomolar ranges, while it rarely affects normal cells [25]. Rigosertib is one of the (E)-styrylbenzylsulfones (I in Figure 1), the biological activity of which is mainly dependent on the nature, position and number of the substituents on the two parent aromatic rings [26]. In phase I/II/III trials, rigosertib has revealed significant therapeutic results patients with solid tumors and hematological malignancies [27,28]. A phase III, open-label, randomized, controlled study of “Rigosertib versus physician’s choice of treatment in MDS (myelodysplastic syndrome) patients after failure of an HMA (hypomethylation agents)” is currently in the stage of recruiting volunteers (registered as NCT02562443).

The coumarin (benzo-α-pyrone) skeleton, a common but important pharmacophore, can interact noncovalently with various active sites, which brings about the various biological activities of coumarins and their derivatives. Therefore, benzopyrone has been widely used as a structural subunit for the discovery of anti-cancer agents [29,30,31,32].

In an attempt to pursue new antitumor agents with better anticancer activities and higher selectivity, the target compounds were developed by molecular modification of the styrylbenzyl sulfones structure (I in Figure 1). A non-classic bioisosteric strategy [33] was used to design new sulfone structures by replacing styryl with a coumarin unit (Figure 1). Then, a series of new coumarin substituted benzylsulfone derivatives (II in Figure 1) were synthesized and assayed with the enzymatic activities against PI3Ks by ELISA, using rigosertib as a comparison. Furthermore, their inhibitory activities against five cancer cell lines were evaluated by MTS ([3-(4,5-dimethylthiazol-2-yl)-5-(3-carboxymethoxyphenyl)-2-(4-sulfophenyl)-2H-tetrazolium, inner salt]) assay. The most active candidates **5h** and **5m** were also screened by wound-healing assay and found to delay Hela cell migration clearly. In addition, the structure–activity relationships of the synthesized compounds and possible enzyme binding modes of **5h** and **5m** were also illustrated.

## 2. Results and Discussion

### 2.1. Chemistry

The target compounds **5a**–**5o** were synthesized via a three-step synthetic route from substituted benzylchloride/bromide (**1a**–**1j**) as outlined in Scheme 1. The starting materials **1a**–**1j** were treated with mercaptoacetic acid in the presence of sodium hydroxide to give benzylmercaptoacetic acids **2a**–**2j** [34], which were oxidized by 30% hydrogen peroxide to give benzylsulfonylacetic acids **3a**–**3j** [35]. Finally, the target compounds (**5a**–**5o**) were synthesized from knoevenagel reaction [36,37] of **3a**–**3j** with substituted salicylaldehydes (**4a**–**4c**) in the presence of DCC (1,3-di-cyclohexylcarbodiimide) and DMAP (4-dimethylaminopyrid). All the target compounds were purified by recrystallization and their structures were confirmed by ^1^H-NMR, ^13^C-NMR and HRMS spectra analysis.

### 2.2. Biological Evaluation

#### 2.2.1. In vitro PI3K Inhibitory Assay

The 15 newly synthesized compounds **5a**‒**5o** were first evaluated for PI3K inhibitory activity in cellular level by ELISA assay [38], using rigosertib as a comparison. Based on the results displayed in Table 1, compounds **5h** and **5m** exhibited the most potent activities with inhibition rates of 50.3% and 50.8% at concentration of 20 μM, respectively, which are comparable to that of rigosertib (53.2%). At 10 μM, **5h** and **5m** also showed moderate inhibitory effects, comparable to rigosertib. These valuable results suggested that **5h** and **5m** could be potent PI3K inhibitors and worthy of further investigation.

#### 2.2.2. Cytotoxicity against Tumor Cells

To investigate the relationship between anticancer activity and PI3K inhibitory activity, the cytotoxicity of these compounds was primarily screened and evaluated by MTS assay in vitro according to the method of Munikrishnappa [39], working with five cancer cell lines (human cervical carcinoma cell line Hela, human liver carcinoma cell line HepG2, human non-small cell lung cancer cell line H1299, human colorectal cancer cell line HCT-116, and human breast carcinoma cell line MCF-7), and rigosertib was used as a reference compound. The in vitro antiproliferative activities are summarized and presented in Table 2. Each IC_50_ (mean ± SD) has been derived from at least three experiments in duplicate. From the recorded IC_50_ values, it was clear that all of the tested compounds showed favorable antitumor activities in low micromolar ranges. In addition, each active compound revealed better activities against Hela and HepG2 when compared to those against other tumor cells. Notably, compounds **5h** and **5m** exhibited remarkable antitumor activities with IC_50_ values of 18.1–32.6 μM and 29.3–42.1 μM, respectively. Furthermore, **5h** was the most effective compound against HepG2 and MCF-7, scoring IC_50_ values of 18.1 μM and 20.5 μM. Nevertheless, other derivatives exhibited only moderate activity against H1299 and HCT-116.The analysis results of antiproliferation activities against tumor cell lines of **5h** and **5m** were in accordance with those of their PI3K inhibitory activities, which suggested that the potent anticancer activities of **5h** and **5m** were likely related to their PI3K inhibitory activities.

From the above data, some interesting structure–activity relationships can be observed: (i) the nature of substituent on the R_2_ position seemed to play an important role for the cytotoxic activity: nitro group > bromine atom > hydrogen atom. For example, in the case of the HepG2 cell line, compounds **5g** and **5h,** which linked with the nitro group, displayed better anticancer activities compared to **5k** and **5m**, which linked with the bromine atom; (ii) the different mono-substituents R_1_ of the benzyl group showed different effects on the anti-tumor activity: fluorine atom (**5f**) > bromine atom (**5e**) > chlorine atom (**5d**); (iii) moreover, the positions of R_1_ also had an evident effect on the activity: para-substitution (**5h**) > ortho-substitution (**5f**) > meta-substitution (**5g**). In short, when R_1_ was para-fluorine and R_2_ was the nitro group, compound (**5h**) showed the best antitumor activity.

#### 2.2.3. Wound-Healing Assays of **5h** and **5m**

Wound-healing assays are some of the earliest and most common techniques to study collective cell migration in vitro due to their simplicity and ability to mimic cell migration during wound healing in vivo [40,41]. The conventional scratch assay is commonly used to measure cell repair rate and provides a simple method to study the directional cell migration in vitro [42,43]. 

Compounds **5h** and **5m**, with the most potent antitumor activity, were further assessed to determine their inhibitory effects on cell migration. Migration of Hela cells was observed at 12 and 24 h after wounding (Figure 2). Hela cells treated with different concentrations of **5h** and **5m** (10 μM and 5 μM) showed different migration rates (Table 2). In the presence of **5h** and **5m**, the wound closure rate of Hela cells was much lower compared to the control group, which suggested that **5h** and **5m** could obviously retard and inhibit Hela cell migration in vitro.

### 2.3. Molecular Docking Studies of 5h and 5m

Docking studies for the synthesized compounds **5h** and **5m**, which showed the most potent inhibitory activity against PI3K, were carried out using the AutoDock 4.2.6 protocol in order to show their binding modes and suggest the possible mechanism of their antiproliferative activity. As PI3K proteins were extracted from Hela cells, PI3Kα and PI3Kβ, which are universally expressed in all cells [12,13], became the focus of this study.

First of all, the validation of the docking protocol was done by re-docking the native ligand into the active sites of PI3Kα (PDB ID: 3HHM [44]) and PI3Kβ (PDB ID: 2Y3A [45]) with the root-mean-square deviations (RMSD) of 0.78 Å and 0.80 Å, respectively (Figure 3). These lower values of RMSD represented the reliability and accuracy of the docking procedure in reproducing the experimentally observed results. Results of the Gibbs free energies (ΔG values) of **5h** and **5m** with PI3Kα and PI3Kβ are summarized in Table 3, which reveal a rough correlation between their binding free energy and their activity.

#### 2.3.1. Binding Modes of Different Compounds with PI3Kα (3HHM)

Figure 4, Figure 5 and Figure 6 demonstrate compounds rigosertib, **5h**, and **5m** docking into the binding site of PI3Kα. Docking results showed that the tested compounds formed interactions with the key amino acid residues such as LYS802, VAL851, ILE932 and ILE848 at its active site. In particular, compound **5h** showed the lowest binding free energy of −8.47 kcal/mole, as compared to rigosertib and **5m** (Table 4). The binding model of compound **5h** into PI3Kα revealed several molecular interactions thought to be responsible for the observed affinity: (i) two hydrogen bond interactions between the two oxygen atoms of the sulfonyl group and VAL851 and SER854; (ii) pi–alkyl and pi–sigma interactions between the benzopyrone ring and ILE932 and ILE848; and (iii) other weak interactions, including C–H bonds, and Van der Waals.

Rigosertib and **5m** formed more hydrogen bonds with PI3Kα than **5h**, five and four respectively. Nevertheless, both rigosertib and **5m** showed higher binding free energies of −7.19 and −7.01. This might be due to the formation of more efficacious hydrogen bonds with the active site. Furthermore, compared with the binding model of rigosertib and **5m** (Figure 4), compound **5h** interacted additionally with SER854. Thus, it seemed that SER854 played a vital role in the binding affinity of the ligand.

#### 2.3.2. Binding Modes of Different Compounds with PI3Kβ (2Y3A)

Figure 7, Figure 8 and Figure 9 demonstrate compounds rigosertib, **5h**, and **5m** docking into the binding site of PI3Kβ. Similarly, compound **5h** showed the lowest binding energy of −7.74. Some molecular interactions in Figure 8 were considered to be responsible for the observed affinity: (i) five hydrogen bond interactions between the ligand and LYS799, TYR833, TRP781 and SER851; (ii) pi–alkyl and pi–sigma interactions between aromatic rings and ILE797, ILE930, ILE845, MET920 and VAL847; and (iii) other weak interactions, including pi–anion, pi–pi T shaped bonds and Van der Waals.

With five hydrogen bond interactions between the ligand and PI3Kβ, rigosertib displayed higher binding energy of −7.26, which is an interesting phenomenon worthy of serious consideration. Finally, it was observed that compound **5h** showed additional bindings to TYR833 and SER851 through hydrogen bond interactions. Another strange example was compound **5m**. There were no hydrogen bond interactions between **5m** and PI3Kβ, but **5m** exhibited binding energy value comparable to that of rigosertib. The halogen interactions, and pi–alkyl and pi–sigma interactions might contribute to this observed affinity.

The nice binding models of compound **5h** and **5m** with PI3K were consistent with the kinase assay data, which indicated that **5h** and **5m** might be potent PI3K inhibitors, especially targeting PI3Kα and PI3Kβ.

## 3. Materials and Methods

### 3.1. Chemistry

All the reagents and solvents were obtained from commercial suppliers (Beijing Innochem Technology Co., Ltd, Beijing, China) and used without further purification, unless otherwise stated. Rigosertib was synthesized and characterized by our research group using the reported method [26]. Reactions were monitored by thin layer chromatography (TLC) on pre-coated silica gel F254 plates with a UV indicator. Melting points were determined on a Uniscience Melting Point apparatus and were uncorrected. ^1^H-NMR and ^13^C-NMR spectra were obtained with a Bruker AM 400 and AM 500 MHz spectrometer (Palo Alto, CA, USA). Mass spectral data were obtained using electron spray ionization on a Micromass ZabSpec high-resolution mass spectrometer (Karlsruhe, Germany). The chemical shifts are reported in parts per million (δ) downfield using tetramethysilane (Me_4_Si) as the internal standard. Spin multiplicities are given as s (singlet), d (doublet), m (multiplet) and q (quartet). Coupling constants (*J* values) were measured in hertz (Hz). 

Note: Only the synthesis and characterization of target compounds are presented in this article. The intermediates mentioned in Scheme 1 are described in the Appendix A.

#### Synthesis of The Target Compounds (5a–5o)

*3-[(4-Methylbenzyl)sulfonyl]-2H-chromen-2-one* (**5a**). To an ice-cold solution of 2-[(4-methylbenzyl)sulfonyl]acetic acid (**3i**) (0.98 g, 4.3 mmol, 1.0 eq.) in THF was added salicylaldehyde (**4a**) (0.55 g, 4.5 mmol, 1.05 eq.), DCC (0.97 g, 4.7 mmol, 1.1 eq.) and DMAP (0.05 g, 0.4 mmol, 0.1 eq.). Stirring was continued at room temperature for 1h, after which time TLC showed the completion of reaction. The precipitate was filtered and the filtrate was concentrated in vacuo almost to dryness, then 50 mL ethyl acetate was added. The transparent solution was washed with dilute hydrochloric acid (20 mL × 4) and saturated brine (20 mL × 2) and dried with anhydrous sodium sulfate. The dried solution was concentrated to get the crude product, which was recrystallized in ethyl acetate to give a white product. Yield 37%, white solid; m.p. 168–170 °C; ^1^H-NMR (400 MHz, DMSO-*d*_6_) δ: 2.25 (s, 3H, -CH_3_), 4.78 (s, 2H), 7.18 (dd, 4H, *J* = 8.0 Hz, Ar-H), 7.46 (t, 1H, *J* = 7.2 Hz, Ar-H), 7.56 (d, 1H, *J* = 8.4 Hz, Ar-H), 7.83 (t, 1H, *J* = 8.4 Hz, Ar-H), 8.01 (d, 1H, *J* = 7.8 Hz, Ar-H), 8.75 (s, 1H, =CH-). ^13^C-NMR (125 MHz, DMSO-*d*_6_) δ: 21.2, 58.7, 117.1, 117.7, 125.0, 125.8, 125.9, 129.7, 131.3, 131.6, 136.1, 138.7, 149.8, 155.2, 156.3. HRMS-ESI (*m/z*): calcd. for C_17_H_14_O_4_S [M + H]^+^ 315.0691, found: 315.0688.

*3-[(4-Bromobenzyl)sulfonyl]-2H-chromen-2-one* (**5b**). Yield 39%, white solid; m.p. 213–214 °C; ^1^H-NMR (400 MHz, DMSO-*d*_6_) δ: 4.84 (s, 2H, -CH_2_-), 7.29 (d, 4H, *J* = 8.4 Hz, Ar-H), 7.47 (t, 1H, *J* = 7.2 Hz, Ar-H), 7.55–7.59 (m, 2H, Ar-H), 7.83 (t, 1H, *J* = 7.6 Hz, Ar-H), 8.02 (d, 1H, *J* = 7.8 Hz, Ar-H), 8.78 (s, 1H, =CH-). ^13^C-NMR (125 MHz, DMSO-*d*_6_) δ: 58.4, 117.0, 117.7, 122.8, 125.6, 126.0, 127.6, 131.7, 132.1, 133.7, 136.2, 150.1, 155.3, 156.2. HRMS-ESI (m/z): calcd. for C_16_H_11_BrO_4_S [M + H]^+^ 378.9640, found: 378.9634.

*3-{[2-(Trifluoromethyl)benzyl]sulfonyl}-2H-chromen-2-one* (**5c**). Yield 37%, pale yellow solid; m.p. 172–174 °C. ^1^H-NMR (400 MHz, DMSO-*d*_6_) δ: 5.02 (s, 2H, -CH_2_-), 7.48 (t, 1H, *J* = 7.6 Hz, Ar-H), 7.57 (d, 1H, Ar-H), 7.63 (t, 1H, *J* = 7.2 Hz, Ar-H), 7.70-7.75 (m, 2H, Ar-H), 7.80-7.87 (m, 2H, Ar-H), 8.06 (d, 1H, *J* = 7.8 Hz, Ar-H), 8. 87 (s, 1H, =CH-). ^13^C-NMR (125 MHz, DMSO-*d*_6_) δ: 55.9, 117.0, 117.9, 125.5, 125.9, 126.6, 127.1 (q, *J* = 5 Hz), 129.3, 129.6, 130.0, 131.8, 133.0, 134.9, 136.2, 149.8, 155.3, 156.3. HRMS-ESI (*m/z*): calcd. for C_17_H_11_F_3_O_4_S [M + H]^+^ 369.0408, found: 369.0403.

*3-[(2-Chlorobenzyl)sulfonyl]-6-nitro-2H-chromen-2-one* (**5d**). Yield 7%, pale yellow solid; m.p. 258.5–260 °C. ^1^H-NMR (400 MHz, DMSO-*d*_6_) δ: 5.01 (s, 2H, -CH_2_-), 7.40–7.43 (m, 2H, Ar-H), 7.50 (d, 1H, *J* = 2.0 Hz, Ar-H), 7.57 (d, 1H, *J* = 2.4 Hz, Ar-H), 7.79 (d, 1H, *J* = 8.8 Hz, Ar-H), 8.59 (d, 1H, *J*_1_ = 2.8 Hz, *J*_2_ = 9.2 Hz, Ar-H), 8.99 (s, 1H, =CH-), 9.04 (d, 1H, *J* = 2.8 Hz, Ar-H). ^13^C-NMR (125 MHz, DMSO-*d*_6_) δ: 56.5, 118.2, 118.7, 126.2, 127.4, 128.0, 128.3, 130.2, 131.3, 134.5, 135.0, 144.5, 149.1, 155.5, 158.7. HRMS-ESI (*m*/*z*): calcd. for C_16_H_10_ClNO_6_S [M + Na]^+^ 401.9815, found: 401.9813.

*3-[(2-Bromobenzyl)sulfonyl]-6-nitro-2H-chromen-2-one* (**5e**). Yield 13%, yellow solid; m.p. 252–254 °C. ^1^H-NMR (400 MHz, DMSO-*d*_6_) δ: 5.01 (s, 2H, -CH_2_-), 7.34 (t, 1H, *J* = 8.0 Hz), 7.44 (t, 1H, *J* = 7.6 Hz, Ar-H), 7.56 (d, 1H, *J* = 7.6 Hz, Ar-H), 7.67 (d, 1H, *J* = 7.6 Hz, Ar-H), 7.79 (d, 1H, *J* = 9.2 Hz, Ar-H), 8.60 (d, 1H, *J*_1_ = 2.8 Hz, *J*_2_ = 9.0 Hz, Ar-H), 8.99 (s, 1H, =CH-), 9.05 (d, 1H, *J* = 2.8 Hz). ^13^C-NMR (125 MHz, DMSO-*d*_6_) δ: 58.9, 118.3, 118.7, 125.8, 127.4, 128.0, 128.3, 128.5, 130.2, 131.5, 133.5, 134.5, 144.5, 149.1, 155.5, 158.8. HRMS-ESI (*m*/*z*): calcd. for C_16_H_10_BrNO_6_S [M + Na]^+^ 445.9310, found: 445.9309.

*3-[(2-Fluorobenzyl)sulfonyl]-6-nitro-2H-chromen-2-one* (**5f**). Yield 17%, yellow solid; m.p. 257–259 °C. ^1^H-NMR (400 MHz, DMSO-*d*_6_) δ: 4.90 (s, 2H, -CH_2_-), 7.24 (d, 2H, *J* = 7.2 Hz, Ar-H), 7.43 (t, 1H, Ar-H), 7.49 (t, 1H, Ar-H), 7.79 (d, 1H, *J* = 9.2 Hz, Ar-H), 8.60 (d, 1H, *J*_1_ = 2.8 Hz, *J*_2_ = 9.2 Hz, Ar-H), 8.98 (s, 1H, =CH-), 9.03 (d, 1H, *J* = 2.8 Hz, Ar-H). ^13^C-NMR (125 MHz, DMSO-*d*_6_) δ: 52.9, 115.2 (d, *J* = 15 Hz), 116.1 (d, *J* = 21 Hz), 118.2, 118.7, 125.3 (d, *J* = 3 Hz), 127.4, 127.9, 130.2, 131.9 (d, *J* = 8 Hz), 134.1 (d, *J* = 3 Hz), 144.5, 149.1, 155.4, 158.7, 161.5 (d, *J* = 246 Hz). HRMS-ESI (*m*/*z*): calcd. for C_16_H_10_FNO_6_S [M + H]^+^ 364.0291, found: 364.0290.

*3-[(3-Fluorobenzyl)sulfonyl]-6-nitro-2H-chromen-2-one* (**5g**). Yield 23%, yellow solid; m.p. 232–233 °C; ^1^H-NMR (400 MHz, DMSO-*d*_6_) δ: 4.89 (s, 2H, -CH_2_-), 7.19–7.26 (q, 3H, Ar-H), 7.39–7.45 (q, 1H, Ar-H), 7.78 (d, 1H, *J* = 9.2 Hz, Ar-H), 8.59 (d, 1H, *J*_1_ = 2.8 Hz, *J*_2_ = 9.2 Hz, Ar-H), 9.00 (s, 1H, =CH-), 9.03 (d, 1H, *J* = 2.8 Hz, Ar-H). ^13^C-NMR (125 MHz, DMSO-*d*_6_) δ: 58.6, 116.2, 116.4, 118.3 (d, *J* = 16 Hz), 118.6 (d, *J* = 18 Hz), 127.4, 127.8, 127.9 (d, *J* = 3 Hz), 130.1, 130.3 (d, *J* = 9 Hz), 131.1 (d, *J* = 9 Hz), 144.5, 149.2, 155.5, 158.7, 162.4 (d, *J* = 243 Hz). HRMS-ESI (*m*/*z*): calcd. for C_16_H_10_FNO_6_S [M + Na]^+^ 386.0111, found: 386.0116.

*3-[(4-Fluorobenzyl)sulfonyl]-6-nitro-2H-chromen-2-one* (**5h**). Yield 28%, yellow solid; m.p. 258.5–260 °C. ^1^H-NMR (400 MHz, DMSO-*d*_6_) δ: 4.85 (s, 2H, -CH_2_-), 7.22 (t, 2H, *J* = 8.8 Hz, Ar-H), 7.42 (t, 2H, *J* = 7.0 Hz, Ar-H), 7.80 (d, 1H, *J* = 9.2 Hz, Ar-H), 8.59 (d, 1H, *J*_1_ = 2.4 Hz, *J*_2_ = 9.2 Hz, Ar-H), 8.97 (s, 1H, =CH-), 9.03 (d, 1H, *J* = 2.8 Hz, Ar-H). ^13^C-NMR (125 MHz, DMSO-*d*_6_) δ: 58.3, 116.1 (d, *J* = 21 Hz), 118.2, 118.6, 124.0 (d, *J* = 3 Hz), 127.4, 127.8, 130.1, 133.8 (d, *J* = 9 Hz), 144.5, 149.1, 155.5, 158.5, 163.0 (d, *J* = 244 Hz). HRMS-ESI (*m*/*z*): calcd. for C_16_H_10_FNO_6_S [M + Na]^+^ 386.0111, found: 386.0109.

*6-Nitro-3-{[2-(trifluoromethyl)benzyl]sulfonyl}-2H-chromen-2-one* (**5i**). Yield 21%, pale yellow solid; m.p. 255–256 °C. ^1^H-NMR (400 MHz, DMSO-*d*_6_) δ: 4.98 (s, 2H, -CH_2_-), 7.59–7.65 (t, 1H, Ar-H), 7.69–7.72 (t, 2H, Ar-H), 7.74–7.81 (q, 2H, Ar-H), 8.57 (d, 1H, *J*_1_ = 2.8 Hz, *J*_2_ = 9.2 Hz, Ar-H), 9.04 (s, 1H, Ar-H), 9.03 (s, 1H, =CH-). ^13^C-NMR (125 MHz, DMSO-*d*_6_) δ: 56.1, 118.4, 118.6, 125.6, 125.5, 127.1 (q, *J* = 6 Hz), 127.4, 128.7, 129.5 (q, *J* = 30 Hz), 130.0, 130.1, 133.1, 134.9, 144.5, 148.9, 155.5, 158.8. HRMS-ESI (m/z): calcd. for C_17_H_10_F_3_NO_6_S [M + Na]^+^ 436.0079, found: 436.0076.

*6-Bromo-3-[(2-fluorobenzyl)sulfonyl]-2H-chromen-2-one* (**5j**). Yield 37%, white solid; m.p. 223–224 °C.^1^H-NMR (400 MHz, DMSO-*d*_6_) δ: 4.89 (s, 2H, -CH_2_-), 7.19–7.26 (q, 2H, Ar-H), 7.40–7.50 (m, 2H, Ar-H), 7.55 (d, 1H, *J* = 8.9 Hz, Ar-H), 7.99 (d, 1H, *J*_1_ = 2.2 Hz, *J*_2_ = 8.6 Hz, Ar-H), 8.28 (d, 1H, *J* = 2.2 Hz, Ar-H), 8.73 (s, 1H, =CH-). ^13^C-NMR (125 MHz, DMSO-*d*_6_) δ: 52.8, 115.4 (d, *J* = 15 Hz), 116.0, 116.1, 117.4, 119.4 (d, *J* = 26 Hz), 125.2 (d, *J* = 3 Hz), 126.9, 131.8 (d, *J* = 8 Hz), 133.4, 134.1 (d, *J* = 3 Hz), 138.4, 148.8, 154.4, 155.7, 161.5 (d, *J* = 246 Hz). HRMS-ESI (*m*/*z*): calcd. for C_16_H_10_BrFO_4_S [M + H]^+^ 398.9525, found: 398.9523.

*6-Bromo-3-[(3-fluorobenzyl)sulfonyl]-2H-chromen-2-one* (**5k**). Yield 20%, white solid; m.p. 233–235 °C. ^1^H-NMR (400 MHz, DMSO-*d*_6_) δ: 4.87 (s, 2H, -CH_2_-), 7.16–7.24 (m, 3H, Ar-H), 7.38–7.43 (m, 1H, Ar-H), 7.54 (d, 1H, *J* = 8.8 Hz, Ar-H), 7.98 (d, 1H, *J*_1_ = 2.4 Hz, *J*_2_ = 8.9 Hz, Ar-H), 8.28 (d, 1H, *J* = 2.4 Hz, Ar-H), 8.75 (s, 1H, =CH-). ^13^C-NMR (125 MHz, DMSO-*d*_6_) δ: 58.6, 116.2 (d, *J* = 20 Hz), 117.4, 118.3 (d, *J* = 21 Hz), 119.32, 119.58, 126.9, 127.8 (d, *J* = 3 Hz), 130.5 (d, *J* = 8 Hz), 131.1 (d, *J* = 9 Hz), 133.4, 138.3, 148.8, 154.4, 155.8, 162.3 (d, *J* = 243 Hz). HRMS-ESI (*m/z*): calcd. for C_16_H_10_BrFO_4_S [M + H]^+^ 398.9525, found: 398.9520.

*6-Bromo-3-[(3-methylbenzyl)sulfonyl]-2H-chromen-2-one* (**5l**). Yield 46%, white solid; m.p. 206–207 °C. ^1^H-NMR (400 MHz, DMSO-*d*_6_) δ: 2.75 (s, 3H, -CH_3_), 4.77 (s, 2H, -CH_2_-), 7.11 (d, 1H, *J* = 7.6 Hz, Ar-H), 7.16 (s, 1H, Ar-H), 7.18 (s, 1H, Ar-H), 7.25 (t, 1H, *J* = 7.8 Hz, Ar-H), 7.54 (d, 1H, *J* = 8.8 Hz, Ar-H), 7.98 (d, 1H, *J*_1_ = 2.4 Hz, *J*_2_ = 8.8 Hz, Ar-H), 8.28 (d, 1H, *J* = 2.4 Hz, Ar-H), 8.74 (s, 1H, =CH-). ^13^C-NMR (125 MHz, DMSO-*d*_6_) δ: 21.3, 59.0, 117.4, 119.3, 119.6, 127.1, 127.6, 128.6, 129.0, 130.0, 132.2, 133.3, 138.2, 138.3, 148.6, 154.3, 155.9. HRMS-ESI (m/z): calcd. for C_17_H_13_BrO_4_S [M + H]^+^ 394.9776, found: 394.9772.

*6-Bromo-3-[(4-fluorobenzyl)sulfonyl]-2H-chromen-2-one* (**5m**). Yield 49%, white solid; m.p. 224–225 °C. ^1^H-NMR (400 MHz, DMSO-*d*_6_) δ: 4.83 (s, 2H), 7.18–7.23 (t, 2H, Ar-H), 7.37–7.41 (t, 2H, Ar-H), 7.54 (t, 1H, *J* = 8.8 Hz, Ar-H), 7.98 (d, 1H, *J*_1_ = 2.4 Hz, *J*_2_ = 8.8 Hz, Ar-H), 8.28 (d, 1H, *J* = 2.4 Hz, Ar-H), 8.73 (d, 1H, =CH-). ^13^C-NMR (125 MHz, DMSO-*d*_6_) δ: 58.2, 116.1 (d, *J* = 21 Hz), 117.4, 119.3, 119.6, 124.2 (d, *J* = 3 Hz), 126.9, 133.4, 133.7 (d, *J* = 8 Hz), 138.3, 148.8, 154.3, 155.9, 162.9 (d, *J* = 244 Hz). HRMS-ESI (*m/z*): calcd. for C_16_H_10_BrFO_4_S [M + H]^+^ 396.9545, found: 396.9538.

*6-Bromo-3-[(2-bromobenzyl)sulfonyl]-2H-chromen-2-one* (**5n**). Yield 27%, white solid; m.p. 233–234 °C. ^1^H -NMR (400 MHz, DMSO-*d*_6_) δ: 5.0 (s, 2H, -CH_2_-), 7.33 (t, 1H, *J* = 7.6 Hz, Ar-H), 7.43 (t, 1H, *J* = 7.6 Hz, Ar-H), 7.52–7.56 (t, 2H, Ar-H), 7.66 (d, 1H, *J* = 8.0 Hz, Ar-H), 7.99 (d, 1H, *J*_1_ = 2.4 Hz, *J*_2_ = 8.8 Hz, Ar-H), 8.29 (d, 1H, *J* = 2.4 Hz, Ar-H), 8.75 (s, 1H, =CH-). ^13^C-NMR (125 MHz, DMSO-*d*_6_) δ: 58.8, 117.3, 119.3, 119.7, 125.8, 127.4, 128.1, 128.5, 131.4, 133.4, 133.4, 134.5, 138.3, 148.8, 154.4, 155.8. HRMS-ESI (*m/z*): calcd. for C_16_H_10_Br_2_O_4_S [M + H]^+^ 458.8724, found: 458.8718.

*6-Bromo-3-[(3-bromobenzyl)sulfonyl]-2H-chromen-2-one* (**5o**). Yield 19%, white solid; m.p. 235–236 °C. ^1^H-NMR (400 MHz, DMSO-*d*_6_) δ: 4.86 (s, 2H, -CH_2_-), 7.33 (s, 1H, Ar-H), 7.34 (s, 1H, Ar-H), 7.53–7.59 (m, 3H, Ar-H), 7.99 (d, 1H, *J*_1_ = 2.4 Hz, *J*_2_ = 8.8 Hz, Ar-H), 8.29 (d, 1H, *J* = 2.4 Hz, Ar-H), 8.77 (s, 1H, =CH-). ^13^C-NMR (125 MHz, DMSO-*d*_6_) δ: 58.4, 117.4, 119.3, 119.6, 122.1, 126.9, 130.5, 130.7, 131.2, 132.1, 133.4, 134.3, 138.3, 148.9, 154.4, 155.8. HRMS-ESI (*m/z*): calcd. for C_16_H_10_Br_2_O_4_S [M + H]^+^ 458.8724, found: 458.8718.

### 3.2. Biological Evaluation

#### 3.2.1. PI3K Inhibitory Activity Assay

The PI3K inhibitory activity assay was performed as described in the literature [36]. The ELISA kit for PI3K (purchased from Shanghai Biological Technology Inc., Shanghai, China) was used according to the manufacturer’s instructions. Hela cells were incubated with DMSO (vehicle control) and different concentrations (10 μM and 20 μM) of rigosertib, and compounds **5h** and **5m**, for 12 h separately. Total cell lysates were immunoprecipitated with anti-PI3K antibody, and PI3K activity was assessed by ELISA. Finally, the kinase assay was detected at 450 nm with microplate reader and the inhibition rate was calculated using the following equation [46].

Inhibition rate (%) = (OD_negative control_ − OD_compounds_)/(OD_negative control_ − OD_blank_) × 100%.

#### 3.2.2. Cytotoxic Activity Assays

Four cell lines (Hela, HepG2, H1299 and MCF-7) were cultured using Dulbecco’s Modified Eagle’s medium (DMEM), and HCT-116 was cultured in RPMI1640 medium. Both media were supplemented with 10% FBS (fetal bovine serum) and 1% penicillin and streptomycin (*v*/*v*). Tumor cells were seeded in 96-well tissue culture plates at a density of 4000–6000 cells/mL at 37 °C in a humidified, 5% CO_2_ atmosphere for 24 h. Then pre-set concentrations of tested compounds were added into 3 wells. After 48 h of incubation, the supernatant was replaced by fresh medium (100 μL/well), and 10 μL MTS reagent (some tetrazolium inner salt) was added to each well. After another 2 h of incubation at 37 °C, the optical density was measured at a wavelength of 492 nm on an ELISA microplate reader. The inhibition rates were calculated by the formula in Section 3.2.1. Next, IC_50_ values for the tested compounds on each cell line were calculated by nonlinear regression analysis using IBM SPSS Statistics v20.0 (IBM-SPSS, Chicago, IL, USA).

#### 3.2.3. Wound Healing Assays

First, 6-well culture plates with markers on the outside bottom of the plates were used as reference points during image acquisition. Then, Hela cells were seeded into plates at a density of 1–1.3 × 10^6^ per well in DMEM medium with 10% FBS. After the cells formed a 100% confluent monolayer, a 20-μL pipette tip was used to generate a wound by manually scraping, which was vertical to the parallel line marked on the plates. The cells were washed thrice with 1 mL of PBS to remove the cell debris from scratches. Next, 2 mL of DMEM with 1% FBS or 2 mL DMEM with 1% FBS and certain concentrations of the tested compounds (10 μM and 5 μM) were added to each well.

Images were obtained using an Olympus IX71 microscope (Olympus, Richmond Hill, ON, Canada) with 10× objective lenses based on the markers, which was in favor of comparing images of each sample at different time points. The wound area was quantified by Image-Pro Plus 6.0 software (Media Cybernetics, Silver Spring, MD, USA). The migration of cells toward the wounds was displayed as a percentage of wound closure: wound closure (%) = [(A_t = 0 h_ − A_t = Δ h_)/A_t = 0 h_] × 100%, where, A_t = 0 h_ is the area of the wound measured immediately after scratching, and A_t = Δ h_ is the area measured 12 or 24 h later [47].

#### 3.2.4. Molecular Docking Studies

Molecular docking of compounds **5h** and **5m** into the three-dimensional X-ray structures of PI3Kα (PDB code: 3HHM) and PI3Kβ (PDB code: (2Y3A) was carried out using AutoDock (version 4.2.6). The three-dimensional structures of compounds **5h** and **5m** were built using ChemBio 3D Ultra 12.0 software (Cambridge Soft Corporation, USA (2009)), which were then energetically minimized by using MMFF94. The protein crystal structures of PI3Kα and PI3Kβ were retrieved from the RCSB Protein Data Bank (http://www.rcsb.org). All ligands and bound waters were eliminated from the protein and the polar hydrogen was added. Docking parameters were set to default values, but only the generated conformations for each ligand were increased to 100. All docked poses of **5h** and **5m** were clustered using a tolerance of 2 Å for RMSD and were ranked in light of the binding docking energies. The desired conformation with lowest energy in the most populated cluster was selected for the following study. Discovery Studio 2017 R2 software was used to perform the visualization of the docking results [48].

## 4. Conclusions

In summary, to obtain effective lead compounds with new structures that can serve as anti-tumor agents, 15 coumarin substituted benzylsulfone derivatives **5a**–**5o** were designed, synthesized, fully characterized and evaluated. It was concluded that most of these derivatives exhibited potent activities against enzymatic potencies and cell proliferation in vitro. In particular, compounds **5h** and **5m** displayed the most potent inhibitory activity against PI3K with inhibition rates of 50.3% and 50.8% at 20 μM, respectively. In addition, **5h** and **5m**, with broad spectrum antitumor activities, showed the greatest inhibitory activities against five tumor cell lines with IC_50_ values of 18.1–32.6 μM and 29.3–42.1 μM, respectively. Moreover, **5h** and **5m** could significantly inhibit the migration of Hela cells. The following in silico molecular docking investigations of compounds **5h** and **5m** into PI3Kα and PI3Kβ indicated that they could fit appropriately into active sites of PI3Kα and PI3Kβ, and interact well with important residues. The above results reveal that compounds **5h** and **5m** could be possible lead compounds against PI3K in cancer therapies and are worthy of further study and optimization.

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
