# Peer review of "Design, Synthesis and Preliminary Biological Evaluation of Benzylsulfone Coumarin Derivatives as Anti-Cancer Agents"

_molecules, 2019, doi:10.3390/molecules24224034_

Round 1
Reviewer 1 Report
This is a solid study on the synthesis and biological activity of a series of coumarin derivatives. The syntheses are quite routine. A series of benzyl halides were reacted with mercaptoacetic acid to give thioether linked carboxylic acids that were oxidized to the corresponding sulfones. DCC mediated esterification with salicylaldehydes, followed by spontaneous cyclization, afforded the coumarin products. These structures resemble rigidified vinyl sulfones such as rigosertib and hence have potential value as anticancer agents. The coumarins were tested for kinase inhibition and in vitro studies were performed on several tumor cell lines. Some significant, if not exceptional, activity was observed. Some structure-activity relationships were explored but the variations in activity were often not all that large. Some modeling studies were also performed.
The new compounds are mostly well characterized but I do have some reservations about the carbon-13 NMR data. I assume that the authors are reporting proton decoupled spectra (13C{1H}), in which case it is not necessary to designate the peaks as singlets. In fact, this is rather misleading. Having said this, compounds containing fluorine will show coupling because 19F is an NMR active nucleus (I = 1/2). So when the compounds contain fluorine, I would expect some of the carbon resonances to be coupled. Unfortunately, the reported data does not follow my expectations. For 5c, which contains a trifluoromethyl group, I would expect coupling to give rise to quartets, not doublets. However, two doublets are reported. Compound 5e does not contain fluorine but is said to have a doublet at 128.41 ppm. Why would this arise? Compound 5i contains a CF3 unit and gives a quartet as 129.52 ppm. However, the authors also report three doublets. Why would there be any doublets for this structure? 5l does not contain fluorine but is said to have a doublet at 138.32 ppm. How is this possible? The same goes for 5n which is reported to have a doublet at 133.49 ppm but should not have any heteronuclear coupling. And finally, a doublet is reported for 5o even though there is no reason to expect any coupling. The authors need to provide a detailed explanation for the reported values. As these descriptions appear to be in error, I am not sure that any of the spectroscopic results can be trusted.
The writing is mostly reasonable but could be improved in places. Contractions such as "What's" (line 66) are not acceptable. Minor grammatical corrections would be helpful. e.g. line 63 should be "to design new sulfone structures"; line 77, "starting materials"; line 81 "in the presence"; etc. In the experimental, line 228 should be "quartet" and "t" should be defined.
If the authors can explain the weird spectroscopic results, the work is publishable.
Author Response
Dear Editors and Reviewer:
Thank you for your letter and for the reviewers’ comments concerning our manuscript entitled “Design, Synthesis and Preliminary Biological Evaluation of Benzylsulfone Coumarin Derivatives as Anticancer Agents” (ID: molecules-626553). Those comments are all valuable and very helpful for revising and improving our paper, as well as the important guiding significance to our researches. We have studied comments carefully and have made correction which we hope meet with approval. Revised portion are marked in red in the paper. The main corrections in the paper and the responds to the reviewer’s comments are as flowing:
Response to Reviewer 1 Comments
Reviewer #1: This is a solid study on the synthesis and biological activity of a series of coumarin derivatives. The syntheses are quite routine. A series of benzyl halides were reacted with mercaptoacetic acid to give thioether linked carboxylic acids that were oxidized to the corresponding sulfones. DCC mediated esterification with salicylaldehydes, followed by spontaneous cyclization, afforded the coumarin products. These structures resemble rigidified vinyl sulfones such as rigosertib and hence have potential value as anticancer agents. The coumarins were tested for kinase inhibition and in vitro studies were performed on several tumor cell lines. Some significant, if not exceptional, activity was observed. Some structure-activity relationships were explored but the variations in activity were often not all that large. Some modeling studies were also performed.
Point 1: The new compounds are mostly well characterized but I do have some reservations about the carbon-13 NMR data. I assume that the authors are reporting proton decoupled spectra (13C{1H}), in which case it is not necessary to designate the peaks as singlets. In fact, this is rather misleading. Having said this, compounds containing fluorine will show coupling because 19F is an NMR active nucleus (I = 1/2). So when the compounds contain fluorine, I would expect some of the carbon resonances to be coupled. Unfortunately, the reported data does not follow my expectations. For 5c, which contains a trifluoromethyl group, I would expect coupling to give rise to quartets, not doublets. However, two doublets are reported. Compound 5e does not contain fluorine but is said to have a doublet at 128.41 ppm. Why would this arise? Compound 5i contains a CF3 unit and gives a quartet at 129.52 ppm. However, the authors also report three doublets. Why would there be any doublets for this structure? 5l does not contain fluorine but is said to have a doublet at 138.32 ppm. How is this possible? The same goes for 5n which is reported to have a doublet at 133.49 ppm but should not have any heteronuclear coupling. And finally, a doublet is reported for 5o even though there is no reason to expect any coupling. The authors need to provide a detailed explanation for the reported values. As these descriptions appear to be in error, I am not sure that any of the spectroscopic results can be trusted. If the authors can explain the weird spectroscopic results, the work is publishable.
Response 1: We are very sorry for the trouble caused by these weird results. In the previous work, 13C-NMR spectra of all the target compounds were analyzed and calculated by MestReNova software, which led to some mistakes and weird results, but unfortunately we did not find them. Thanks to your valuable comments and opinions, we recheck and re-analyze the 13C-NMR data carefully this time. Finally, we get lots of reasonable results, which are consistent with your expectations. Indeed, compound 5c and compound 5i, which contain trifluoromethyl groups, give quartets. 5c gives a quartet at 127.1 ppm and no doublets. 5i gives two quartets at 127.1 ppm and 129.5 ppm, and no doublets. Compound 5f, which contains fluorine, gives six doublets. The same goes for 5g, 5j and 5k. Compounds 5h and 5m, which contain fluorine, give four doublets. Compounds 5a, 5b, 5d, 5e, 5l, 5n, and 5o, which do not contain fluorine, give no doublet. This phenomenon is related to the substituted position of fluorine atom on the aromatic ring. Furthermore, all of the NMR data is rechecked and corrected. Details were presented in the revised manuscript.
Point 2: The writing is mostly reasonable but could be improved in places. Contractions such as "What's" (line 66) are not acceptable. Minor grammatical corrections would be helpful. e.g. line 63 should be "to design new sulfone structures"; line 77, "starting materials"; line 81 "in the presence"; etc. In the experimental, line 228 should be "quartet" and "t" should be defined.
Response 2: We are very sorry for our incorrect writing, and we have made correction according to Reviewer’s comments. Furthermore, this manuscript has undergone English language editing by MDPI to improve the language. Details were presented in the revised manuscript.
We tried our best to improve the manuscript and made some changes in the manuscript. We appreciate for Editors/Reviewers’ warm work earnestly, and hope that the correction will meet with approval. Once again, thank you very much for your comments and suggestions.
Reviewer 2 Report
The manuscript describes the synthesis of a library of 15 benzylsulfone coumarin derivatives in an efficient 3 step synthesis. The compounds are screened for inhibitory activity in a P13K ELISA assay, and for cytotoxicity in 5 cancer cell lines in another assay. All compounds gave god inhibitory effects on scale with the standard compound Rigosertib, an anticancer agent currently in phase III testing. The cytotoxicity results gave much weaker results than Rigosertib, but the same two componds 5h and 5m were the best in both thests. These two compounds were further studied for inhibitory effect in a cell migration assay, and in silico docking studies into receptors P13Kα and P13Kβ giving binding energies similar to Rigosertib.
The manuscript is well written and the introduction is well referenced. Actually, 34 of 42 references belongs to the introduction. Obviously the use of literature in the following discussions is limited.
In the synthesis the first two reactions are referenced, but the key Koevenagel cyclization reaction lacks a reference, but there is no indication that this cyclization is novel to this paper. Thus references lack here.
Another problem that has to be addressed is the unconscious use of number of decimals. The manuscript consistently and blindly uses two decimals after the decimal-point, no matter accuracy of numbers, reasonable accuracy of tests or anything. This must be corrected, and the numbers must reflect the accuracy of measurements:
Table 1: apparently 1 repetition only. Stating 53.22+/- 3.66 make no sense. The real variation here is 53+/- 4 There is no real value in giving these %-number with decimals at all.
Table 2: Three replicates, so these micromolar concentrations should be more accurate that % in table 1. Still they add 10 microliter of a reagent. To get the claimed accuracy they would have to add 10.00 microliter. Maybe the number of decimals in the table should be reduced.
NMR: 125 MHz: 13C is given with 2 decimals. This is highly unusual, as it is generally accepted that 13C is only given with one decimal accuracy.
Yields in the experimental descriptions (In the list of products a line of which starting materials were combined to make them should be added) yields are given with 1 decimal. Usually yields are given as whole % unless the experiment is really large scale or repeated several times with small deviation in yield.
A lot of data is given in SI. The SI was not available for review!
I lack the necessary competence to further evaluate the docking studies, but they appear sound, and the conclusion is in agreement with the data.
Author Response
Dear Editors and Reviewers:
Thank you for your letter and for the reviewers’ comments concerning our manuscript entitled “Design, Synthesis and Preliminary Biological Evaluation of Benzylsulfone Coumarin Derivatives as Anticancer Agents” (ID: molecules-626553). Those comments are all valuable and very helpful for revising and improving our paper, as well as the important guiding significance to our researches. We have studied comments carefully and have made correction which we hope meet with approval. Revised portion are marked in red in the paper. The main corrections in the paper and the responds to the reviewer’s comments are as following:
Response to Reviewer 2 Comments
Reviewer #2: The manuscript describes the synthesis of a library of 15 benzylsulfone coumarin derivatives in an efficient 3 step synthesis. The compounds are screened for inhibitory activity in a P13K ELISA assay, and for cytotoxicity in 5 cancer cell lines in another assay. All compounds gave god inhibitory effects on scale with the standard compound Rigosertib, an anticancer agent currently in phase III testing. The cytotoxicity results gave much weaker results than Rigosertib, but the same two componds 5h and 5m were the best in both thests. These two compounds were further studied for inhibitory effect in a cell migration assay, and in silico docking studies into receptors P13Kα and P13Kβ giving binding energies similar to Rigosertib.
Point 1: The manuscript is well written and the introduction is well referenced. Actually, 34 of 42 references belong to the introduction. Obviously the use of literature in the following discussions is limited.
Response 1: According to the Reviewer’s suggestion, six other related references have been used in the discussions. Details were presented in the revised manuscript.
Point 2: In the synthesis the first two reactions are referenced, but the key Koevenagel cyclization reaction lacks a reference, but there is no indication that this cyclization is novel to this paper. Thus references lack here.
Response 2: According to the Reviewer’s suggestion, two references about Koevenagel cyclization reaction have been cited in the article. Details were presented in the revised manuscript.
Point 3: Another problem that has to be addressed is the unconscious use of number of decimals. The manuscript consistently and blindly uses two decimals after the decimal-point, no matter accuracy of numbers, reasonable accuracy of tests or anything. This must be corrected, and the numbers must reflect the accuracy of measurements:
Table 1: apparently 1 repetition only. Stating 53.22+/- 3.66 make no sense. The real variation here is 53+/- 4. There is no real value in giving these %-number with decimals at all.
Response 3: We are very sorry for our incorrect writing. Because each data in Table 1 comes from the result of 3 duplicates calculated by the formula in Section 3.2.1, we think that it is necessary to give the %-numbers one decimal accuracy. Thus, 53.22 ± 3.66 should be 53.2 ± 3.7. The same goes for other data in Table 1. Details were presented in the revised manuscript.
Point 4: Table 2: Three replicates, so these micromolar concentrations should be more accurate that % in table 1. Still they add 10 microliter of a reagent. To get the claimed accuracy they would have to add 10.00 microliter. Maybe the number of decimals in the table should be reduced.
Response 4: According to the Reviewer’s suggestion, the number of decimals in Table 2 has been reduced to one. Furthermore, the number of decimals in Table 3 also has been reduced to one. Details were presented in the revised manuscript. As for rigosertib (the reference compound), the IC50 values will seem too small when expressed in micromotor. So, we give the IC50 values of rigosertib two decimals accuracy. Details were presented in the revised manuscript.
Point 5: NMR: 125 MHz: 13C is given with 2 decimals. This is highly unusual, as it is generally accepted that 13C is only given with one decimal accuracy.
Response 5: According to the Reviewer’s suggestion, the chemical shifts of 13C have been given one decimal accuracy and the coupling constants have not been given any decimals. Details were presented in the revised manuscript.
Point 6: Yields in the experimental descriptions (In the list of products a line of which starting materials were combined to make them should be added) yields are given with 1 decimal. Usually yields are given as whole % unless the experiment is really large scale or repeated several times with small deviation in yield.
Response 6: According to the Reviewer’s suggestion, yields in the experimental descriptions have been given as whole %. Details were presented in the revised manuscript.
Point 7: A lot of data is given in SI. The SI was not available for review!
Response 7: The supporting information (SI) and manuscript has been submitted in the MDPI system at the same time. So we have no idea why the SI was not available for review. But that's okay, since we upload the SI here again. Please find and check it in the attachment. We regret any inconvenience caused.
Point 8: I lack the necessary competence to further evaluate the docking studies, but they appear sound, and the conclusion is in agreement with the data.
Response 8: Much thanks to you for your good comments.
We tried our best to improve the manuscript and made some changes in the manuscript. We appreciate for Editors/Reviewers’ warm work earnestly, and hope that the correction will meet with approval. Once again, thank you very much for your comments and suggestions.
